# Diagnosis of Secondary Bacterial Meningitis via Aromatic Metabolites and Biomarkers in Cerebrospinal Fluid

**DOI:** 10.3390/ijms262110522

**Published:** 2025-10-29

**Authors:** Petr A. Meinarovich, Ekaterina A. Sorokina, Natalia V. Beloborodova, Alisa K. Pautova

**Affiliations:** Federal Research and Clinical Center of Intensive Care Medicine and Rehabilitology, Petrovka Str. 25-2, 107031 Moscow, Russia; pmeynarovich@fnkcrr.ru (P.A.M.); esorokina@fnkcrr.ru (E.A.S.); nbeloborodova@fnkcrr.ru (N.V.B.)

**Keywords:** nosocomial meningitis, post-neurosurgical meningitis, chronic critical illness, microbial metabolites, aromatic metabolites, 4-hydroxyphenyllactic acid, phenyllactic acid, indole-3-lactic acid, interleukin-6, multivariate prognostic model

## Abstract

The development of sensitive and specific diagnostic methods for secondary bacterial meningitis remains an urgent challenge in neurosurgical and intensive care units. A combination of various clinical and biochemical parameters, as well as biomarkers and metabolites in cerebrospinal fluid (CSF), can be considered for constructing multivariate diagnostic models. In this study, 96 CSF samples from 53 patients with suspected secondary meningitis were analyzed. The first cohort, consisting of patients with sequelae of severe brain damage, included 7 patients (21 CSF samples) with and 29 patients (56 CSF samples) without secondary bacterial meningitis. The second cohort comprised patients after neurosurgical interventions, including 10 patients (12 CSF samples) with and 7 patients (7 CSF samples) without secondary bacterial meningitis. Combined group 1 with 33 CSF samples from patients with secondary bacterial meningitis and combined group 2 with 63 CSF samples from patients without secondary bacterial meningitis had statistically different cell and biochemical compositions and higher CSF concentrations of biomarkers (interleukin-6 and S100 protein) and lactate-containing aromatic metabolites in group 1. Univariate prognostic models constructed on 4-hydroxyphenyllactic, phenyllactic, and indole-3-lactic acids demonstrated outstanding AUC-ROC of more than 0.91. A multivariate model built on all biomarkers and metabolites resulted in AUC-ROC = 0.94 with a sensitivity of 0.94 and specificity of 0.86, and was found to be the most accurate method for the diagnosis of secondary bacterial meningitis.

## 1. Introduction

The diagnosis and monitoring of secondary meningitis remain a challenging and urgent issue in neurosurgical and intensive care units [1,2]. During the management of meningitis, cerebrospinal fluid (CSF) samples may be collected longitudinally for bacteriological, clinical, and biochemical analysis. Although the gold standard for confirming the presence of meningitis is a positive CSF culture or the identification of a pathogen by polymerase chain reaction (PCR), false-negative results can occur due to ongoing antibiotic therapy, which is often administered to treat other infectious complications, thus complicating the objective diagnosis and monitoring of secondary meningitis [3]. Various clinical and biochemical parameters routinely assessed during CSF testing include the evaluation of pleocytosis and its cellular composition, as well as glucose and protein concentrations [4,5]. Occasionally, CSF lactate or procalcitonin analysis has also been introduced into routine practice. However, previous studies have demonstrated the suboptimal diagnostic accuracy of these parameters [6,7]. In addition, the underlying pathology that leads to the development of secondary meningitis most likely plays a vital role in determining the threshold values of these diagnostic parameters [8].

Severe infectious complications disrupt numerous metabolic pathways in the patient’s body. From a diagnostic perspective, altered metabolism of the proteinogenic aromatic amino acids—tyrosine, phenylalanine, and tryptophan—is of interest, as some metabolites of these amino acids are of microbial origin [9,10], and their profile changes with the development of infectious complications. These metabolites have been studied as prognostic and diagnostic markers of local and systemic infectious processes for more than two decades. In healthy individuals, the source of low concentrations of circulating aromatic microbial metabolites is the gut microbiota. With the development of severe infectious complications and sepsis, the microbial metabolism of aromatic amino acids changes and occurs not only in the gut, but also at sites of infection. Intermediate or alternative metabolites enter the bloodstream in excess, leading, in particular, to increased levels of circulating sepsis-associated metabolites [9,10,11].

One of our studies focused on exploring their potential utility in post-neurosurgical meningitis diagnosis [12]. Phenylalanine and tyrosine metabolites, specifically phenyl-containing acids, were detected in the CSF of patients with suspected secondary meningitis via gas chromatography–mass spectrometry (GC-MS). This method demonstrated limited sensitivity, resulting primarily in the detection of only 4-hydroxyphenyllactic acid (*p*-HPhLA) in the CSF samples, with its statistically higher concentration in the CSF samples of patients with suspected secondary meningitis (*p* < 0.001), threshold value of 0.9 µmol/L, and moderate receiver operating characteristic (ROC) analysis parameters including an area under the curve (AUC) of 0.73, sensitivity of 0.67, and specificity of 0.83 [12]. In a recent study conducted on patients with sequelae of severe brain damage, a broader profile of aromatic compounds, including indole-containing tryptophan metabolites, was quantified in serum and CSF samples via ultra-high-performance liquid chromatography–tandem mass spectrometry (UPLC-MS/MS) [13]. This method exhibited excellent sensitivity several orders of magnitude greater than GC-MS, allowing for the quantitative determination of a wide range of metabolites [14]. However, the limited number of analyzed samples from a small patient cohort precluded the construction of robust prognostic models with appropriate threshold values [13].

Moreover, in a study on post-neurosurgical patients [12], we studied some specific biomarkers in CSF, such as interleukin-6 (IL-6, proinflammatory marker) [15], neuron-specific enolase (NSE) [16], and protein S100 [17] (markers of brain cell damage). The univariate predictive model constructed on IL-6, which was the only biomarker with a statistically higher concentration in the CSF samples of patients with suspected secondary meningitis (*p* < 0.001), demonstrated a perfect sensitivity of 0.96 with a poor specificity of 0.54 and a moderate AUC-ROC of 0.78 (threshold value = 270 pg/mL) [12].

Based on our study predicting postoperative complications in patients after cardiac surgery via various uni- and multivariate models [18], we concluded that when univariate models did not demonstrate ideal performance characteristics, it was reasonable to construct multivariate models that combined serum metabolites with various biomarkers and clinical and biochemical data. Such an integrated approach ultimately led to building models with satisfactory predictive characteristics (AUC-ROC = 0.85).

Thus, a combination of CSF *p*-HPhLA with high specificity and CSF IL-6 with high sensitivity, together with some other potentially significant biomarkers and aromatic metabolites on a larger group of patients, may improve the characteristics of predictive models. The present study aimed to develop various prognostic models for secondary bacterial meningitis based on the concentrations of several aromatic metabolites and biomarkers identified in the CSF samples from patients following neurosurgical interventions (post-neurosurgical patients), as well as from patients with sequelae of severe brain damage.

## 2. Results and Discussion

### 2.1. Description of Patients from Cohorts #1 and #2

In the present study, two distinct cohorts of patients with suspected secondary meningitis were enrolled (Table 1): patients with sequelae of severe brain damage (cohort #1) and patients after neurosurgical interventions (cohort #2). The patients in cohort #1 had long-term sequelae of severe brain damage. These patients were admitted to our intensive care units from other hospitals after the acute phase of their condition had passed, but the patient required an extended stay in the intensive care unit. These types of patients are also called patients with a chronic critical illness. This group of patients is characterized by an extended stay in the intensive care unit, in most cases on prolonged mechanical ventilation. Unfortunately, the development of nosocomial infections (pneumonia, secondary meningitis, ventriculitis, urogenital tract infections, and soft tissue infections) is frequent in these patients [19,20]. Cohort #2 included patients after neurosurgical intervention who developed secondary meningitis within the first week of being in the neurointensive care unit. The demographic parameters indicate a male predominance among the cohorts and an age comparability.

In cohort #1, secondary meningitis was diagnosed in 7 patients (group 1.1), and 29 patients did not have secondary meningitis (group 1.2). Most patients from cohort #1 were studied over time. The etiologies leading to severe brain damage included a traumatic brain injury (42%), hemorrhagic events (22%), a stroke (19%), central nervous system (CNS) tumors (14%), and a CNS infection (meningoencephalitis). Some patients (*n* = 24) had pneumonia as an infectious complication that is consistent with previously published data on patients with a chronic critical illness [19]. Mortality was significantly higher in group 1.1 (86%) compared to group 1.2 (10%).

Microbiological analyses were performed on the patients’ CSF. Different nosocomial agents were present among the detected bacteria. The presence of *Cryptococcus neoformans* in one sample most likely indicates that there are bacterial species that are not identified when using the cultural method [21]. *Staphylococcus epidermidis* in samples from patients without meningitis is a typical example of contamination. One sample from group 1.1 had a Gram-positive strain but no growth on culture.

In cohort #2, secondary meningitis was diagnosed in 10 patients (group 2.1), and 7 patients did not have secondary meningitis (group 2.2). The primary etiology leading to neurosurgical interventions included CNS tumors. There were no fatalities and non-CNS infections in this cohort. *Staphylococcus aureus* (MRSA) is the only proven pathogen found using the cultural method [22].

The high mortality in group 1.1 highlights the importance of accurate diagnosis, especially in patients with a chronic critical illness [20], and the need to improve diagnostic strategies to ensure timely intervention and reduce mortality in this high-risk group.

### 2.2. Description of Cerebrospinal Fluid Samples from Cohorts #1 and #2

Since patients from cohorts #1 and #2 differ in the time of development of secondary meningitis relative to the primary brain injury, we considered it appropriate to conduct a comparative analysis between both cohorts to study the CSF composition in neurosurgery patients and patients with chronic critical illness, as well as between groups within and between cohorts (Table 2).

Most patients were studied over time and we used the Wald test (a mixed-effects model) for statistical comparison of two groups to account for the sample belonging to a specific patient.

Significant differences were identified regarding the CSF cellular and biochemical composition between cohorts #1 and #2 (lymphocytes, neutrophils, and protein) as well as between groups within these cohorts in patients with and without secondary meningitis (groups 1.1 and 1.2: leucocyte count, lymphocytes, neutrophils, and protein; groups 2.1 and 2.2: neutrophils and glucose). The CSF samples from patients with secondary meningitis (groups 1.1 and 2.1) did not differ in terms of CSF cellular and biochemical composition, while patients without secondary meningitis (groups 1.2 and 2.2) exhibited statistically significant differences in leucocyte count, lymphocytes, and protein.

The median concentrations of CSF biomarkers (IL-6, NSE, and S100) were higher in cohort #2 compared to cohort #1, with statistical differences in the IL-6 and S100 concentrations for groups 1.1 and 1.2, as well as for groups 1.2 and 2.2. While concentrations of IL-6 in all groups and cohorts were significantly higher than the reference value of 1.5 pg/mL from one study [23], concentrations of NSE were within the reference range of 17.3 ± 4.6 ng/mL [24] in cohort #1 and higher in group 2.2 of cohort #2. Generally, the concentrations of S100 in cohort #1 were not much higher than the reference value of 1.4 μg/L [25]; however, the concentrations in group 1.1 were clearly above the reference. The concentrations of S100 in cohort #2 were obviously higher than the reference values in all groups.

A set of eleven aromatic metabolites was quantified in the CSF samples. However, two phenyl-containing metabolites, PhPA and *p*-HPhPA, were not detected in most samples and were omitted from Table 2. Aromatic metabolites in the CSF samples from patients from cohorts #1 and #2 were not statistically different, and their median concentrations were generally comparable. However, CSF samples from patients with secondary meningitis (group 1.1) in cohort #1 had statistically higher concentrations of *p*-HPhLA, *p*-HPhAA, PhLA, 3ILA, and 3ICA in comparison with group 1.2 without secondary meningitis, which is consistent with the data we obtained in our previous study on a smaller number of patients with a chronic critical illness [13]. Moreover, the concentrations of 3ICA in CSF were not previously statistically higher in patients with secondary meningitis.

Despite higher median concentrations of *p*-HPhLA, PhLA, 3ILA, 5HIAA, and 3IPA in the CSF samples of patients with secondary meningitis from cohort #2 (group 2.1) compared to group 2.2, these changes were not statistically significant. This may be due to the specific statistical method of comparison together with a relatively small number of samples in groups 2.1 and 2.2 (12 and 7 samples, respectively). Previously, on a larger cohort of post-neurosurgical patients (*n* = 82), we obtained statistically higher *p*-HPhLA concentrations in CSF samples from patients with suspected secondary meningitis [12].

There were no statistically significant differences between the CSF samples from patients with secondary meningitis in both cohorts (groups 1.1 and 2.1); however, the median concentrations of *p*-HPhLA, *p*-HPhAA, and PhLA were higher in group 1.1. Finally, the median concentrations of aromatic metabolites were comparable in patients without secondary meningitis in both cohorts (groups 1.2 and 2.2), with median concentrations of 3ILA, 3IAA, and 3IPA statistically being higher in group 2.2 compared to group 1.2.

Additionally, it is possible to compare the concentrations of most indole-containing metabolites, except 3ICA, and *p*-HPhLA, with some reference values from published data. First of all, it should be noted that the presented reference values are below the lower limit of quantitation (LLOQ) of the UPLC-MS/MS method used [14]; therefore, we can only discuss the results obtained above the corresponding LLOQs. The concentrations of *p*-HPhLA were higher than its LLOQ (37.5 nmol/L) in all CSF samples, thus significantly exceeding the reference range from a study on healthy people [26]. Concentrations of *p*-HBA, *p*-HPhAA, PhLA, and 3ICA were also higher than their corresponding LLOQs. The concentrations of 5HIAA were higher than the LLOQ (20 nmol/L) in all groups, except group 1.2. The concentrations of 3ILA and 3IAA were higher than the LLOQs (2 and 10 nmol/L, respectively) in all groups except group 1.2. The concentrations of 3IPA were both above and below the LLOQ (2 nmol/L) in all groups.

### 2.3. Cerebrospinal Fluid Composition from the Combined Sample Groups of Patients with or Without Secondary Meningitis

At the main stage, we combined CSF samples from patients with secondary meningitis from both cohorts into group 1, and CSF samples from patients without secondary meningitis from both cohorts into group 2, and compared them (Table 3).

All standard CSF parameters measured during routine laboratory CSF testing were statistically different between the two groups. CSF leukocyte count showed a profound statistically significant increase in group 1, with a median of 434 cells/mm^3^ compared to 7 cells/mm^3^ in group 2. The percentage of CSF lymphocytes was lower in group 1 than in group 2 (10% versus 74%, *p* < 0.001), confirming a shift away from lymphocytic toward neutrophilic predominance. Correspondingly, CSF neutrophils were 89% in group 1 versus 37% in group 2 (*p* < 0.001), characterizing bacterial inflammation. Secondary bacterial meningitis cases uniformly exhibited raised CSF protein, with a median concentration of 2.6 g/L, significantly higher than in group 2 (median 0.7 g/L, *p* < 0.001). Notably, a shift in CSF cell populations was observed, with a notable decrease in lymphocyte percentage and a corresponding increase in neutrophil predominance—a hallmark of bacterial, rather than viral, meningitis [27,28]. CSF glucose was notably decreased in group 1 than in group 2 subjects (2.1 versus 3.4 mmol/L, *p* < 0.001), reflecting bacterial glucose metabolism. Moreover, all of these CSF cellular and biochemical parameters are comparable with those obtained in our previous study on post-neurosurgical patients [12].

IL-6 is a key cytokine playing a crucial role in the inflammatory processes within the CNS. It is produced by endothelial cells, astrocytes, and glial cells in response to different types of injury, and it also participates in neurogenesis. IL-6 stimulates the production of acute-phase proteins and can lead to disruption of the blood–brain barrier [29]. The CSF IL-6 levels were significantly elevated in group 1, with a median of 3228 pg/mL, showing a marked inflammatory cytokine response in patients with secondary meningitis. At the same time, although we observed lower concentrations of 61 pg/mL in group 2, these concentrations were still much higher than the reference value of 1.5 pg/mL (Table 2), indicating the presence of an inflammatory process, albeit less intense than in group 1. However, when comparing these results with those obtained in our previous study on post-neurosurgical patients [12], patients without signs of secondary meningitis demonstrated a higher median value of 227 pg/mL. In comparison, patients with signs of secondary meningitis had a relatively comparable median value of 2678 pg/mL.

S100 proteins signal glial and neuronal injury and actively participate in antimicrobial defense and immune activation, including the induction of IL-6, thereby amplifying inflammatory cascades during bacterial infection [30]. The CSF S100 protein levels were statistically elevated in group 1, with median concentrations reaching 4.7 μg/L compared to 0.7 μg/L in group 2, which is within the reference range of 1.4 ± 0.5 (Table 2). This highly significant difference not only reflects the activation of astroglial cells, but also indicates ongoing neuronal injury during secondary meningitis [17]. In our previous study on post-neurosurgical patients [12], S100 concentrations were comparable between the two patient groups with no statistical difference.

Three lactate-containing aromatic microbial metabolites (*p*-HPhLA, PhLA, and 3ILA) showed multiple increases in CSF concentration in group 1 compared to group 2. *p*-HPhLA is a microbial-origin metabolite resulting from tyrosine metabolism and is elevated in patients with secondary meningitis, as demonstrated in our previous studies [12,13]. Generally, its median concentrations in patients with secondary meningitis were comparable in all three studies, similarly to its median concentrations in patients without secondary meningitis in all three studies. This high statistical significance, observed over several studies, highlights that *p*-HPhLA can be a reliable biomarker for the diagnosis of secondary meningitis. Moreover, this metabolite was the only one among other aromatic metabolites quantified in a study on simultaneously collected CSF and serum samples from patients with a chronic critical illness, the concentrations of which were higher in CSF than in the corresponding serum samples, indirectly indicating its microbial origin in the CNS [13].

PhLA and 3ILA are also microbial-origin metabolites resulting from phenylalanine and tyrosine metabolism, respectively, and have been found to be 10 and 20 times higher in patients with secondary meningitis compared to those without secondary meningitis (572 and 156 nmol/L versus 50 and 8 nmol/L, respectively) in our previous study on patients with a chronic critical illness [13]. In a current study, we obtained similar PhLA and 3ILA concentrations in patients without secondary meningitis (52 and 8 nmol/L) and lower concentrations in patients with secondary meningitis (311 and 91 nmol/L).

In order not to repeat data on the previously demonstrated diagnostic significance of phenyl-containing metabolites, we will provide a reference to the book chapter that has accumulated the results of numerous clinical and experimental studies [11]. Information on the pathophysiological and potential diagnostic significance of indole-containing acids is demonstrated in our recent study [13].

### 2.4. Dynamic Changes in CSF Parameters in Patients from Cohort #1

We followed patients over time, since it is important to make a diagnostic decision about the presence or absence of meningitis and to monitor the patient’s condition at every point. Hence, we included not only points at the time of onset of secondary meningitis, but points after antimicrobial treatment started. Our hypothesis was that metabolites and biomarkers could change dynamically, reflecting the patient’s condition, and that these changes would be noticeable in the event of improvement or deterioration. Our hypothesis was based on our previous clinical study on the dynamics of metabolite concentrations in the serum of patients with sepsis, in which we demonstrated not only dynamic changes in metabolite concentrations, but also the synchrony of these changes with the clinical picture of the patients’ condition. We even observed a slight advance in metabolite changes compared to the SOFA scale [31].

To demonstrate the importance of monitoring of the metabolites and biomarkers over time, data for two patients with secondary meningitis and two patients without secondary meningitis from cohort #1 are shown in Figure 1. A small number of culture-positive meningitis in group 1 (only eight patients with a positive CSF culture, Table 1) does not allow us to conduct a correlation analysis between detected bacteria and determined analytes in CSF. However, some information about detected bacteria and antimicrobial treatment for these four patients will be provided. Patient 1 without secondary meningitis had a surgical intervention for a CNS tumor and had no antibacterial treatment. Patient 2 without secondary meningitis had intracranial hemorrhage and received Cefoperason and Sulbactam 2 g + 2 g two times per day as antimicrobial prophylaxis. Patient 3 was admitted to our center with sequelae of traumatic brain injury from another hospital and then secondary bacterial meningitis occurred. There was no growth in culture and secondary meningitis was diagnosed based on the Centers for Disease Control (CDC) and Prevention criteria. Antibacterial treatment for secondary meningitis included Meropenem 1 g three times per day and Linezolid 600 g two times per day for therapy of nosocomial pneumonia caused by *Staphylococcus aureus*. Patient 4 with subarachnoid hemorrhage had nosocomial meningitis caused by *Klebsiella pneumonia* after intracranial pressure monitoring was started. At considered points Imipenem 0.5 g and Cilastatin 0.5 g four times per day were started after stopping of Cefoperason and Sulbactam 2 g + 2 g two times per day and Amikacin in unknown dosage.

In Figure 1a,b, there are no significant changes over time for the metabolites and biomarkers, which were statistically different in Table 3 (IL-6, *p*-HPhLA, PhLA, and 3ILA). Even for *p*-HPhLA in Figure 1b, which seems to have dynamic changes with the highest concentration in point 2, its real concentrations were less than 1050 nmol/L. In contrast, Figure 1c demonstrates synchronic dynamics for leucocyte count in CSF and *p*-HPhLA, while IL-6 and two other lactic metabolites did not show significant changes. In Figure 1d, *p*-HPhLA and leucocyte count in the CSF change in the same direction, while IL-6 shows the opposite dynamics. Since leucocyte count in CSF is commonly used for secondary meningitis diagnosis, we compared its dynamics with those for metabolites and biomarkers. Based on the obtained data, we can conclude that *p*-HPhLA has the most similar dynamic changes with leucocyte count in CSF.

### 2.5. Prognostic Models for Secondary Bacterial Meningitis

At the final stage, we constructed various univariate prognostic models and a multivariate model to predict secondary bacterial meningitis (Table 4). The bootstrapping technique was used to calculate the confidence interval (CI) for every metric in this table.

The diagnostic performance of the CSF cell and biochemical parameters, selected metabolites, and biomarkers was evaluated using ROC analysis [32]. Among routine CSF parameters, leukocyte count (AUC-ROC = 0.91), lymphocyte (AUC-ROC = 0.88), and neutrophil percentage (AUC-ROC = 0.93) showed the highest discriminative ability. CSF protein demonstrated a moderate diagnostic accuracy of AUC-ROC = 0.83, while CSF glucose had a lower AUC-ROC of 0.78. These results align with a study where good but somewhat lower accuracies for peripheral blood leukocytes (AUC-ROC = 0.815) and classic CSF markers such as leukocytosis and glucose levels were reposted [33]. The relatively high predictive ability of these parameters in our study may be explained by the fact that they are used in CDC criteria to classify patients into groups. However, their perfect diagnostic capabilities were previously demonstrated and validated [34].

Among biomarkers, while neuronal markers showed relatively low diagnostic value (NSE: AUC-ROC = 0.71, S100: AUC-ROC = 0.74), IL-6 reached an AUC-ROC of 0.82 with a moderate sensitivity of 0.75 and relatively high specificity of 0.84 at a threshold concentration of 869 pg/mL. In a recent study by other researchers, elevated IL-6 concentrations were found in the CSF of patients with nosocomial meningitis compared to both control subjects and those with pleocytosis not related to meningitis. The diagnostic value of IL-6 in CSF was highlighted by a high specificity of 0.95 and low sensitivity of 0.55 at a threshold level greater than 440 pg/mL [15]. In another study, IL-6 levels were evaluated to distinguish viral and bacterial meningitis, including those with negative culture or not confirmed by PCR. For the diagnosis of bacterial meningitis, IL-6 demonstrated AUC-ROC = 0.937 (95% CI: 0.895–0.978) with 0.95 sensitivity and 0.77 specificity for the threshold value of 1418 pg/mL, and 0.64 sensitivity and 0.97% specificity for the threshold value of 15,060 pg/mL [35]. In a meta-analysis based on 13 studies, CSF IL-6 had the following characteristics for bacterial meningitis diagnosis: 0.91 sensitivity, 0.93 specificity, and AUC-ROC = 0.97 (95% CI 0.95–0.98) [36].

The analysis of metabolites highlighted several strong discriminators. *p*-HPhLA (AUC-ROC = 0.91), PhLA (AUC-ROC = 0.92), and 3ILA (AUC-ROC = 0.91) showed outstanding diagnostic performance, with both high sensitivities (0.79–0.90) and specificities (0.82–0.97). In contrast, *p*-HBA and 3ICA had low AUC-ROC values (0.53 and 0.65, respectively), indicating limited diagnostic utility. Other metabolites, including *p*-HPhAA, 5HIAA, 3IAA, and 3IPA, demonstrated moderate but less robust performance with AUC-ROC values ranging from 0.65 to 0.79.

One multivariate model that was built on metabolites and biomarkers was evaluated, and the resulting metrics were computed. The complex structure of the data (more than one sample per patient) prevented us from using classic machine learning models due to the fact that we must consider the variability of samples from one patient. This is the reason why mixed-effects models were used. The core model of the pipeline was statsmodels.BinomialBayesMixedGLM. The output of this model was used to fit the logistic regression from Sklearn and consider class imbalance. The SHAP method was used for the feature selection and importance evaluation of every single feature, taking into account pairwise correlations (the feature_perturbation parameter was set to ‘correlation_dependent’). The model was evaluated with GroupKfold cross-validation, and features with a SHapley Additive exPlanations (SHAP) value greater than the median of all features were included in the model in each iteration. The present multivariate model, when compared with univariate models, demonstrated the best AUC-ROC score of 0.94 and sensitivity of 0.94, while the specificity was not the best but also very high at 0.86. The results of the model estimation are explained in Figure 2.

The horizontal bar plot (Figure 2a) presents the mean absolute SHAP values for all considered features. The most informative features were *p*-HPhLA, IL-6, and PhLA, each showing substantial contributions to model predictions. 3ICA and 3IAA also had moderate importance, while the importance of features such as *p*-HBA, *p*-HPhAA, 5HIAA, and the encoded Cohort variable was lower. These results highlight the role of specific metabolites and biomarkers in classification performance, supporting their potential value for distinguishing disease groups.

The confusion matrix (Figure 2b) summarizes model prediction performance in terms of true and false positives and negatives. Out of the total samples, 57 true negatives and 27 true positives were identified, alongside 6 false positives and 6 false negatives. This suggests a high degree of correct classification for both groups, with the errors being evenly distributed. The observed balance in the error types supports robust model sensitivity and specificity. 

The ROC curve (Figure 2c) demonstrates the trade-off between sensitivity and specificity across all possible thresholds. The calculated AUC-ROC is 0.94, indicating the excellent discriminative ability of the model. The shaded region represents the 95% CI obtained via the bootstrap method.

Individual metabolites and biomarkers demonstrate notable specificity but limited sensitivity when considered separately (Table 4). For example, markers such as IL-6, *p*-HPhLA, and PhLA show moderate to high specificity (ranging from approximately 0.75 to 0.97) but often lower or more variable sensitivity. Conversely, some metabolites, like 5HIAA and 3ILA, exhibit higher sensitivity but less consistent specificity. In contrast, our multivariate model combining these metabolites and biomarkers improves both sensitivity and specificity simultaneously (0.94 and 0.86, respectively), resulting in the highest AUC-ROC. This integrated approach effectively balances the trade-off between sensitivity and specificity, enhancing overall diagnostic accuracy beyond what is achievable with individual markers alone.

These findings are complementary to prior reports in the literature. A near-perfect diagnostic accuracy for bacterial meningitis was reported using a combination of serum procalcitonin and CSF protein, achieving an AUC-ROC of 1.00 with 1.00 sensitivity and 0.97 specificity [33]. While their individual markers—CSF protein (AUC-ROC = 0.99) and procalcitonin (AUC-ROC = 0.95)—outperformed individual markers in our study, the integration of specialized metabolites in our model (notably *p*-HPhLA, PhLA, and 3ILA with AUC-ROC 0.91–0.92) provided comparably robust individual discriminatory power. This supports the added value of metabolic profiling alongside classical biomarkers in enhancing diagnostic precision.

A combination of biomarkers such as procalcitonin (AUC-ROC = 0.92), CSF neutrophil-to-lymphocyte ratio (AUC-ROC = 0.87), and lactate (AUC-ROC = 0.77) improved the diagnostic accuracy, with the combined AUC-ROC reaching 0.93 [37]. These results closely mirror the performance of our multivariate model, highlighting the consistent advantage of integrated biomarker panels across different cohorts and study designs.

Lastly, recent machine learning applications to meningitis diagnosis [38] have reported AUC-ROC values up to 0.95 using ensemble algorithms such as Random Forest and CatBoost. Our use of SHAP analysis to identify key predictive features—*p*-HPhLA, IL-6, and PhLA—parallels this approach, emphasizing the importance of targeted biomarkers alongside routine clinical parameters for optimized classification.

Taken together, the findings of our study corroborate accumulating evidence that the integration of metabolic data and classical biomarkers within advanced computational frameworks significantly enhances the accuracy of secondary bacterial meningitis diagnosis, offering potential improvements in clinical decision-making, especially in complex and secondary cases where traditional markers may be insufficient. Future work should focus on prospective validation and integration into clinical workflows to establish real-world utility.

Our study has objective limitations. We consider neither bacterial species nor treatment for patients due to the absence of such data. Specification by these features may influence the results of our study. Moreover, a small number of unique patients as a complex hierarchical data structure may contribute to bias and affect the results from the models used. Stricter planning of the study would allow the use of more simple methods for data analysis.

## 3. Materials and Methods

### 3.1. Study Design

The study, which was conducted at the Federal Research and Clinical Center of Intensive Care Medicine and Rehabilitology (Moscow, Russia), included patients with and without signs of secondary (nosocomial) bacterial meningitis (*n* = 53, 35 men, 18 women, aged from 19 to 82, 96 CSF samples) and was approved by the Local Ethics Committee of the Federal Research and Clinical Center of Intensive Care Medicine and Rehabilitology (Protocol excerpt No. 3/24/3 dated 13 November 2024) in accordance with the Helsinki Declaration. Informed consent was obtained from all subjects involved in the study or their legal representatives.

Cohort #1 (*n* = 36, 22 men, 14 women, aged from 22 to 82, 77 CSF samples) consisted of patients with sequelae of severe brain damage with a chronic critical illness, including those with secondary bacterial meningitis (*n* = 7, 3 men, 4 women, aged from 22 to 73, 21 CSF samples) and without secondary bacterial meningitis (*n* = 29, 19 men, 10 women, aged from 23 to 82, 56 CSF samples). Cohort #2 (*n* = 17, 13 men, 4 women, aged from 19 to 71, 19 CSF samples) consisted of patients after neurosurgical interventions, including those with secondary bacterial meningitis (*n* = 10, 7 men, 3 women, aged from 19 to 67, 12 CSF samples) and without secondary bacterial meningitis (*n* = 7, 6 men, 1 woman, aged from 25 to 71, 7 CSF samples).

Most patients from cohort #1 were studied over time (*n* = 19; 10 men and 9 women) and had more than one CSF sample analyzed (1 sample—17 patients, 2 samples—9 patients, 3 samples—3 patients, 4 samples—2 patients, 5 samples—4 patients, 6 samples—1 patient). One patient in cohort #1 had two CSF samples, one categorized into group 1.1 and the other into group 1.2. All samples from a single patient were collected on different days. A detailed allocation of patients to the respective cohorts and groups is shown in Figure 3.

### 3.2. Collection of Clinical Data and Cerebrospinal Fluid Sample Analysis

The clinical and demographic data of the patients, as well as the results of laboratory and biochemical analyses of CSF, and the results of the microbiological analysis of CSF were obtained from medical records.

The diagnosis of secondary bacterial meningitis was retrospectively established in three cases:The patient’s medical records documented the growth of microorganisms in the CSF microbiological examination.There was a positive result from PCR testing.The patient’s medical records included a diagnosis of bacterial meningitis before the sample collection date. They met the CDC criteria [4], with threshold values defined as follows: leukocyte count in CSF greater than 300 cells/mm^3^, protein concentration above 1 g/L, and glucose concentration below 2.8 mmol/L.

CSF samples were collected after a lumbar puncture in patients with suspected secondary meningitis. Following the required clinical testing, the residual CSF samples were frozen at −80 °C for the analysis of biomarkers and metabolites. The concentrations of biomarkers IL-6, NSE, and protein S100 were measured using an electrochemiluminescence method on a Cobas e411 automatic analyzer (Roche, Basel, Switzerland). Metabolite concentrations of 4-hydroxyphenyllactic acid (*p*-HPhLA), 4-hydroxybenzoic acid (*p*-HBA), 4-hydroxyphenylacetic acid (*p*-HPhAA), 3-phenylpropionic acid (PhPA), 4-hydroxyphenylpropionic acid (*p*-HPhPA), 3-phenyllactic acid (PhLA), 5-hydroxyindole-3-acetic acid (5HIAA), indole-3-lactic acid (3ILA), indole-3-carboxylic acid (3ICA), indole-3-acetic acid (3IAA), and indole-3-propionic acid (3IPA) were obtained via a validated UPLC-MS/MS method [14] on a SCIEX ExionLC AC System with AB Sciex Triple Quad 5500 Plus MS (AB Sciex, Framingham, MA, USA).

### 3.3. Statistical Data Analysis

Statistical analysis was conducted to identify features that significantly differed between cohort #1 and cohort #2, as well as between patients without secondary meningitis and those with secondary meningitis within the combined dataset. Data before preprocessing is demonstrated in Appendix A. Some laboratory variables had missing values. The proportion of missing values exceeding 5% was observed for the following parameters: leukocyte count in cerebrospinal fluid (9%), CSF glucose (8%), CSF lymphocytes (14%), CSF neutrophils (17%), and CSF protein (8%). Missing values were imputed using the K-Nearest Neighbors Imputer (KNNImputer) method, excluding the class label from the dataset during imputation to avoid data leakage. No missing values were observed in metabolite and biomarker concentrations. Data were preprocessed with Pandas 2.3.1, Numpy 2.3.2, and Sklearn 1.7.1 [39,40].

Data analysis for our study was performed using mixed-effects models to account for its hierarchical structure (i.e., more than one sample for some patients). Table 5 demonstrates which variables were considered a target variable, fixed, or random effect for every type of analysis. All models were implemented with the Statsmodels 0.14.5 package for Python 3.0 [41].

To compare patients with and without secondary meningitis (as cohort #1 and cohort #2), linear mixed-effects models were used (statsmodels.MixedLM). The studied features were considered as target variables, and the grouping factor was included as a fixed effect. Patient ID was used as a random effect. The Wald test was applied to assess the statistical significance of the model coefficients. For all tests, a *p*-value of less than 0.05 was considered statistically significant. The Benjamini–Hochberg correction was used to account for multiple comparisons [42].

ROC analysis was used to compute the AUC-ROC, sensitivity (a true positive rate), specificity (1—false positive rate), and the optimal threshold value chosen with Jouden’s J statistics to discriminate samples for all studied variables [43]. This analysis was performed using a Generalized Linear Mixed-Effects Model (GLMM) in statsmodels.BinomialBayesMixedGLM. The grouping variable was treated as the target variable, while the studied feature and patient ID were included as fixed and random effects, respectively. To obtain CI of 2.5% and 97.5% percentiles, we used a bootstrap method at the patient level [44].

To classify samples from patients with and without meningitis based on metabolite and biomarker concentrations, we employed a mixed-effects logistic regression model. This model accounts for the hierarchical structure of the data, where multiple samples originate from the same patient, and patients are nested within different clinics. The outcome variable was binary, indicating the presence or absence of meningitis in each sample. Fixed effects included the concentrations of selected metabolites and biomarkers measured in each sample. To properly account for correlations between multiple samples from the same patient, we included a random intercept and random slope for each patient, allowing both the baseline risk and the association between metabolite and biomarker concentration and outcome to vary across individuals. Since the dataset included samples from two cohorts, a cohort number was incorporated as a fixed effect due to the limited number of levels, enabling direct estimation of the cohort-specific differences [45,46].

GroupKFold cross-validation with k = 5 was used, splitting data based on patient identifier (Patient’s_ID). This prevented data from the same patient from entering both the training and test sets, ensuring the proper assessment of generalization to unseen patients. All features were standardized using statistics from each fold’s training set to prevent data leakage (rather than the test set, which would be incorrect). The encoded database variable was always maintained. RandomOverSampler was applied to balance classes in the training set by oversampling the minority class. The primary predictive model was a Bayesian mixed-effects logistic regression (BinomialBayesMixedGLM from statsmodels), which accounts for patient-specific random effects and fixed effects for metabolite features. This structure appropriately accounts for clustering in the data arising from repeated or grouped measurements per patient. Predictions from the main mixed-effects model are used as inputs to a distinct logistic regression model. This two-step approach is used for receiving logits of predictions. SHAP analysis is performed on a surrogate logistic regression model for each training fold to estimate feature importance. Features with absolute SHAP values above the median are selected, focusing the model on the most informative predictors. For each fold, probabilities for the positive class are calculated and concatenated across the entire dataset. The primary evaluation metric is the average AUC-ROC score across all cross-validation folds, providing a robust measure of discrimination. Final feature importance is visualized with a horizontal bar plot showing mean absolute SHAP values (averaged across folds), identifying the most informative features in the model [47].

## 4. Conclusions

In this study, which included patients with and without secondary meningitis, we were able to once again convincingly demonstrate the possibilities of using microbial metabolites as markers of infectious complications, including those in the CNS. Our previous studies on other groups of patients with local (pneumonia and peritonitis) and generalized infectious processes (sepsis and septic shock), in which phenyl-containing microbial metabolites in the blood serum demonstrated their diagnostic and prognostic capabilities, formed the basis for the development of our research in the field of CNS infections. This study on post-neurosurgery patients and patients with a chronic critical illness is generalizing and ultimately offers a whole set of possible prognostic models. Univariate models are the most attractive from the point of view of use in clinical practice. The models we have proposed, based on lactate-containing aromatic acids, demonstrate outstanding characteristics and can be proposed for clinical use, but a more comprehensive model, particularly the one proposed in this paper, would allow for an even more accurate determination of the presence of secondary meningitis, which is so vital in neurosurgery and intensive care units. Today, due to the use of mass spectrometry methods to determine metabolites, it is difficult to disseminate our developments in the clinic widely, but we are looking for such opportunities. In the future, we plan to study the probable mechanisms of accumulation of lactate-containing and other aromatic acids in the CSF to create a solid scientific foundation for the patterns we observe.

## Figures and Tables

**Figure 1 ijms-26-10522-f001:**
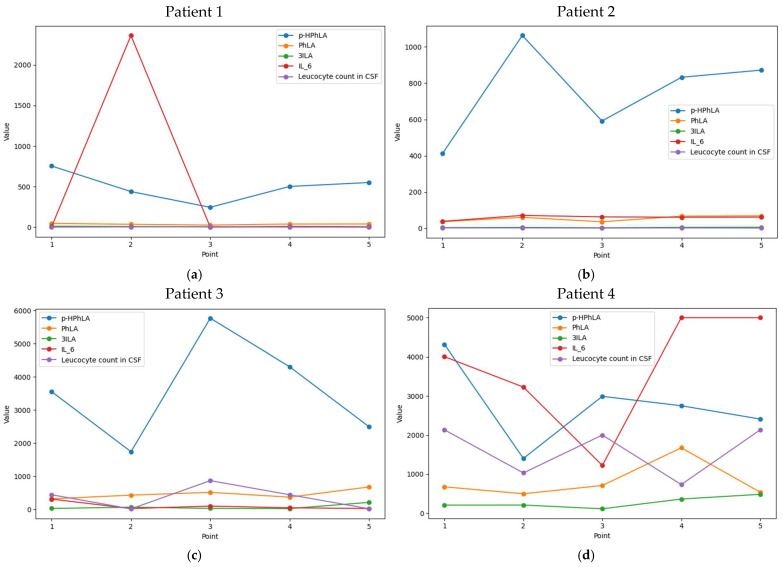
Dynamics of the concentrations of 4-hydroxyphenyllactic (*p*-HPhLA), phenyllactic (PhLA), and indole-3-lactic (3ILA) acids in nmol/L, and interleukin-6 (IL-6) in pg/mL compared to leucocyte count in CSF (cells/mm^3^) for (**a**) Patient 1 and (**b**) Patient 2 without secondary meningitis and for (**c**) Patient 3 and (**d**) Patient 4 with secondary meningitis.

**Figure 2 ijms-26-10522-f002:**
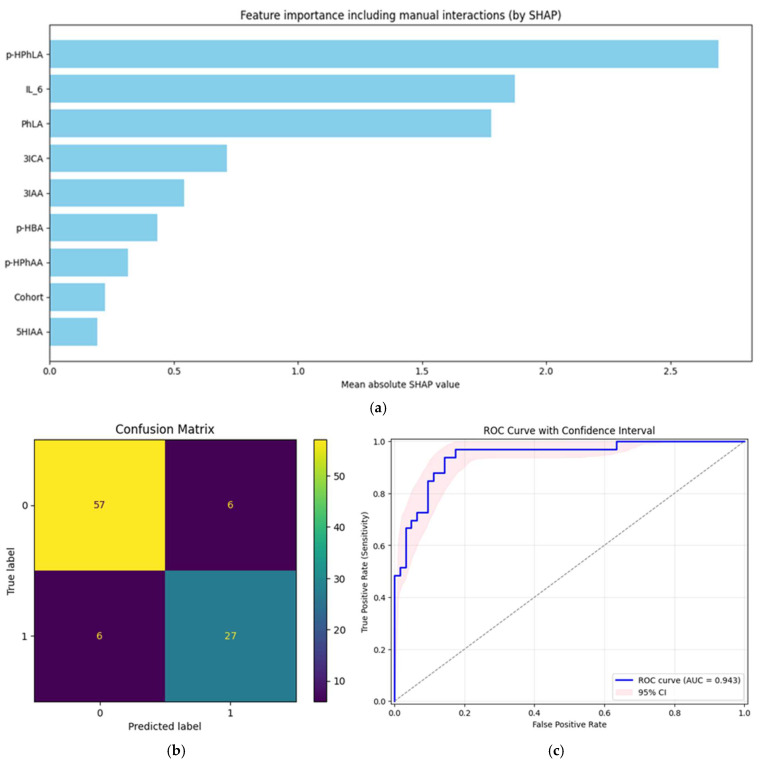
Results of the multivariate model. (**a**) Mean SHAP values for the resulting model; (**b**) confusion matrix demonstrating relations between true and predicted classes; (**c**) ROC curve for the final model with confidence intervals (pink zone).

**Figure 3 ijms-26-10522-f003:**
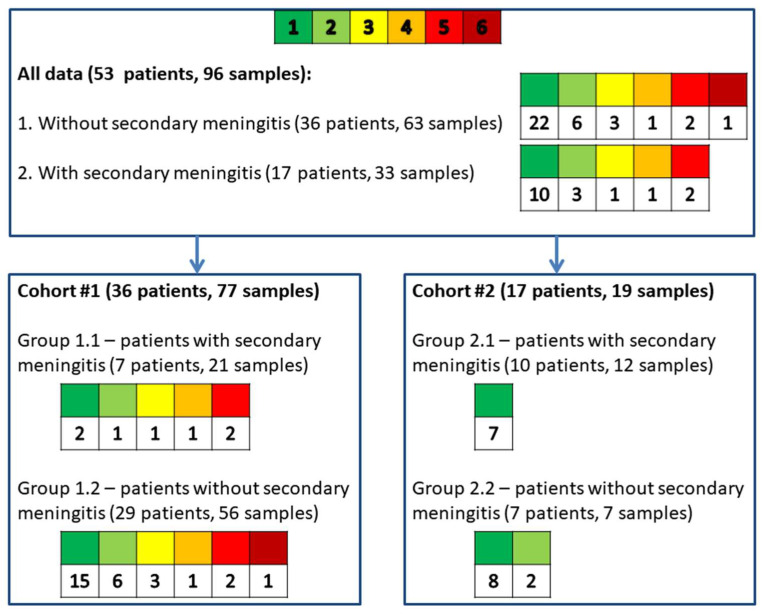
Allocation of patients to the respective cohorts and groups. The colored table demonstrates how many patients correspond to each number of samples—from left green square (explains number of patients with one CSF sample) to right red sample (explains number of patients with six CSF samples).

**Table 1 ijms-26-10522-t001:** Demographic and clinical descriptions of patients with sequelae of severe brain damage (cohort #1) and patients after neurosurgical interventions (cohort #2).

Parameter	Cohort #1		Cohort #2			
	All Patients/Samples	Group 1.1 with Secondary Meningitis	Group 1.2 Without Secondary Meningitis	All Patients/Samples	Group 2.1 with Secondary Meningitis	Group 2.2 Without Secondary Meningitis
Number of patients, *n*	36	7	29	17	10	7
Number of CSF samples, *n*	77	21	56	19	12	7
Sex, males	21	3	19	13	7	6
Age, years	50 (40, 61)	32 (50, 65)	50 (39, 59)	39 (26, 54)	38 (27, 45)	48 (31, 50)
Deaths	9	6	3	0	0	0
Traumatic brain injury	15	1	14	-	-	-
Hemorrhagic events	8	3	5	-	-	-
Ischemic stroke	7	1	6	-	-	-
CNS tumor	5	1	4	17	10	7
CNS infection	1	1	0	-	-	-
Pneumonia	24	5	19	0	0	0
No growth in CSF	29	2	27	15	9	6
*Staphylococcus epidermidis*	2	0	2	1	0	1
*Acinetobacter baumanii*	-	1	-	-	-	-
*Klebsiella pneumonia*	-	2	-	-	-	-
*Cryptococcus neoformans*	-	1	-	-	-	-
*Staphylococcus aureus*	-	0	-	-	1	-
Unknown (positive Gram strain but no growth)	-	1	-	-	-	-

**Table 2 ijms-26-10522-t002:** The cellular and biochemical composition and concentrations of biomarkers and aromatic metabolites in cerebrospinal fluid samples from patients with sequelae of severe brain damage with a chronic critical illness (cohort #1) and patients after neurosurgical interventions (cohort #2), and the results of comparative statistical analysis (the Wald test) between various groups of samples (statistically significant differences are highlighted in bold).

Parameter	Reference Values	Cohort #1				Cohort #2				*p* _Cohorts #1 and #2_	*p* _1.1 and 2.1_	*p* _1.2 and 2.2_
		All Samples (*n* = 77)	Group 1.1 with Secondary Meningitis (*n* = 21)	Group 1.2 Without Secondary Meningitis (*n* = 56)	*p* _1.1 and 1.2_	All Samples (*n* = 19)	Group 2.1 with Secondary Meningitis (*n* = 12)	Group 2.2 Without Secondary Meningitis (*n* = 7)	*p* _2.1 and 2.2_			
Leucocyte count, cells/mm^3^	2–8	11 (3, 96)	733 (139, 2000)	6 (2, 18)	**<0.001**	163 (18, 710)	258 (136, 866)	14 (5, 88)	0.21	0.27	0.62	**<0.001**
Lymphocytes, %	90–95	61 (30, 94)	11 (5, 36)	75 (54, 100)	**<0.001**	4 (3, 6)	4 (3, 8)	4 (3, 5)	0.55	**<0.001**	0.08	**<0.001**
Neutrophils, %	3–5	47 (25, 68)	84 (69, 92)	40 (22, 50)	**<0.001**	88 (44, 96)	96 (87, 97)	37 (11, 63)	**<0.001**	**<0.001**	0.31	0.57
Protein, g/L	0.1–0.3	0.7 (0.5, 1.2)	1.6 (0.6, 4.5)	0.7 (0.5, 1.0)	**<0.001**	2.6 (1.2, 4.2)	3.6 (2.4, 4.7)	1.2 (0.5, 2.2)	0.06	**<0.001**	0.36	**0.03**
Glucose, mmol/L	2.8–3.9	3.1 (2.2, 3.8)	0.6 (0.2, 3.5)	3.2 (2.6, 3.8)	0.12	2.6 (2.2, 3.5)	2.2 (1.9, 2.7)	3.7 (3.3, 4.2)	**<0.001**	0.42	0.99	0.78
IL-6, pg/mL	1.5 (1.0, 2.2) [23]	71 (10, 897)	3228 (104, 5000)	42 (7, 234)	**<0.001**	2336 (420, 5000)	4017 (1455, 5000)	271 (144, 3371)	0.19	**<0.001**	0.46	**0.03**
NSE, ng/mL	17.3 ± 4.6 [24]	1.6 (0.9, 3.2)	2.0 (1.3, 9.3)	1.3 (0.7, 2.9)	0.06	12.1 (5.0, 37.2)	20.5 (9.0, 35.7)	5.0 (3.3, 64.7)	0.7	0.14	0.4	**<0.001**
S100, μg/L	1.4 ± 0.5 [25]	0.8 (0.4, 3.3)	3.7 (0.7, 18.0)	0.6 (0.3, 1.2)	**<0.001**	6.5 (2.5, 23.1)	5.5 (2.3, 26.0)	9.2 (3.2, 19.6)	0.98	**<0.001**	0.7	**<0.001**
*p*-HPhLA, nmol/L	7.8 (5.8, 10.2) [26]	633 (370, 1246)	2750 (1734, 4206)	491 (281, 840)	**<0.001**	925 (533, 1485)	1172 (861, 1726)	522 (424, 725)	0.46	0.7	0.03	0.1
*p*-HBA, nmol/L	no data	33 (23, 61)	35 (23, 99)	32 (22, 60)	0.26	30 (17, 41)	22 (14, 36)	35 (33, 41)	0.38	0.32	0.31	0.7
*p*-HPhAA, nmol/L	no data	223 (101, 469)	906 (362, 3167)	152 (77, 278)	**0.03**	100 (60, 191)	92 (66, 233)	103 (60, 138)	0.7	0.46	0.32	0.79
PhLA, nmol/L	no data	67 (43, 173)	538 (361, 736)	52 (36, 77)	**<0.001**	100 (59, 131)	108 (94, 145)	47 (35, 68)	0.26	0.57	0.16	0.43
5HIAA, nmol/L	1.9 (1.1, 3.6) [26]	112 (22, 224)	162 (85, 306)	72 (<20, 212)	0.35	122 (81, 204)	143 (103, 215)	72 (62, 146)	0.91	0.87	0.77	0.93
3ILA, nmol/L	0.4 (0.3, 0.5) [26]	11 (4, 27)	202 (22, 472)	5 (3, 15)	**<0.001**	25 (18, 85)	60 (25, 93)	10 (6, 18)	0.39	0.95	0.26	**0.03**
3ICA, nmol/L	no data	11 (6, 14)	12 (11, 20)	8 (6, 13)	**0.03**	8 (6, 10)	8 (7, 11)	5 (4, 9)	0.7	0.47	0.17	0.38
3IAA, nmol/L	0.5 (0.4, 1.0) [26]	39 (22, 79)	96 (47, 204)	26 (18, 44)	0.12	71 (43, 118)	73 (65, 88)	47 (35, 177)	0.47	0.14	0.6	**0.03**
3IPA, nmol/L	0.11 ± 0.02 [10]	<2 (<2, 4)	2 (<2, 4)	<2 (<2, 4)	**<0.001**	9 (3, 15)	9 (4, 16)	2 (<2, 14)	0.56	0.27	0.88	**<0.001**

**Table 3 ijms-26-10522-t003:** A comparison of the CSF samples from the combined sample groups of patients with (group 1) and without (group 2) secondary meningitis, and the results of comparative statistical analysis (the Wald test) between various groups of samples (statistically significant differences are highlighted in bold).

Parameter	Group 1. CSF Samples from Patients with Secondary Bacterial Meningitis (*n* = 33)	Group 2. CSF Samples from Patients without Secondary Bacterial Meningitis (*n* = 63)	*p*
Leucocyte count, cells/mm^3^	434 (139, 1685)	7 (2, 18)	**<0.001**
Lymphocytes, %	10 (5, 20)	74 (43, 98)	**<0.001**
Neutrophils, %	89 (77, 96)	37 (20, 50)	**<0.001**
Protein, g/L	2.6 (1.3, 4.7)	0.7 (0.5, 1.1)	**<0.001**
Glucose, mmol/L	2.1 (0.4, 3.0)	3.4 (2.6, 3.9)	**<0.001**
IL-6, pg/mL	3228 (323, 5000)	61 (9, 370)	**<0.001**
NSE, ng/mL	6.3 (1.5, 25.4)	1.7 (0.9, 3.2)	0.1
S100, μg/L	4.7 (1.1, 18.0)	0.7 (0.4, 2.5)	**<0.001**
*p*-HPhLA, nmol/L	1923 (1248, 2991)	502 (306, 828)	**<0.001**
*p*-HBA, nmol/L	30 (19, 58)	34 (23, 58)	0.49
*p*-HPhAA, nmol/L	362 (158, 1938)	149 (75, 271)	0.05
PhLA, nmol/L	311 (108, 668)	52 (36, 76)	**<0.001**
5HIAA, nmol/L	153 (97, 234)	72 (<20, 206)	0.18
3ILA, nmol/L	91 (24, 208)	6 (3, 16)	**<0.001**
3ICA, nmol/L	12 (9, 14)	8 (6, 13)	0.15
3IAA, nmol/L	77 (50, 176)	29 (20, 52)	0.21
3IPA, nmol/L	4 (<2, 10)	<2 (<2, 4)	0.2

**Table 4 ijms-26-10522-t004:** ROC analysis results for the univariate and the multivariate models.

Parameter	AUC-ROC, 95% CI	Sensitivity, 95% CI	Specificity, 95% CI	Threshold Value, 95% CI
Leucocyte count, cells/mm^3^	0.91 (0.84, 0.96)	0.80 (0.67, 1.00)	0.94 (0.71, 1.00)	139 (18, 244)
Lymphocytes, %	0.88 (0.82, 0.94)	0.91 (0.79, 1.00)	0.79 (0.63, 0.92)	48 (16, 68)
Neutrophils, %	0.93 (0.87, 0.98)	0.88 (0.73, 1.00)	0.90 (0.74, 1.00)	58 (46, 81)
Protein, g/L	0.83 (0.75, 0.91)	0.77 (0.68, 0.87)	0.84 (0.72, 0.95)	1.3 (1.0, 2.6)
Glucose, mmol/L	0.78 (0.66, 0.87)	0.71 (0.42, 0.86)	0.82 (0.62, 1.00)	2.6 (1.3, 3.0)
IL-6, pg/mL	0.82 (0.73, 0.92)	0.75 (0.52, 1.00)	0.84 (0.45, 0.96)	869 (16, 5000)
NSE, ng/mL	0.71 (0.59, 0.84)	0.55 (0.38, 0.86)	0.91 (0.53, 0.97)	6.3 (4.6, 24.4)
S100,μg/L	0.74 (0.61, 0.90)	0.71 (0.38, 1.00)	0.75 (0.53, 0.96)	1.7 (0.7, 11.5)
*p*-HPhLA, nmol/L	0.91 (0.84, 0.96)	0.79 (0.65, 0.92)	0.97 (0.84, 1.00)	1248 (924, 1510)
*p*-HBA, nmol/L	0.53 (0.43, 0.65)	0.22 (0.12, 0.95)	0.96 (0.26, 1.00)	180 (14, 466)
*p*-HPhAA, nmol/L	0.70 (0.56, 0.82)	0.67 (0.44, 0.83)	0.75 (0.62, 0.94)	262 (156, 452)
PhLA, nmol/L	0.92 (0.85, 0.97)	0.84 (0.71, 0.95)	0.88 (0.78, 0.98)	99 (78, 173)
5HIAA, nmol/L	0.65 (0.44, 0.75)	0.86 (0.73, 1.00)	0.53 (0.34, 0.70)	79 (69, 178)
3ILA, nmol/L	0.91 (0.85, 0.96)	0.90 (0.74, 1.00)	0.82 (0.70, 0.98)	18 (15, 47)
3ICA, nmol/L	0.65 (0.42, 0.76)	0.81 (0.68, 0.96)	0.56 (0.26, 0.68)	9 (5, 13)
3IAA, nmol/L	0.79 (0.26, 0.88)	0.87 (0.79, 1.00)	0.71 (0.07, 0.85)	45 (37, 332)
3IPA, nmol/L	0.65 (0.36, 0.76)	0.63 (0.33, 1.00)	0.70 (0.02, 0.93)	3 (2, 35)
Multivariate model	0.94 (0.89; 0.98)	0.94 (0.86; 1.00)	0.86 (0.75; 0.96)	-

**Table 5 ijms-26-10522-t005:** Distribution of the variables between corresponding parameters in models.

Part of Analysis	Formula for Each Model		
	Target Variable	Fixed Effects	Random Effects
Statistical comparison of groups (the Wald test)	Concentration of a single metabolite, biomarker, or clinical variable	Group or Cohort	Patient’s ID
ROC analysis	Group	Concentration of a single metabolite, biomarker, or clinical variable	Patient’s ID
Final model	Group	Selected metabolites and biomarkers; Cohort	Patient’s ID

## Data Availability

The original contributions presented in this study are included in the article. Further inquiries can be directed to the corresponding author.

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
