# Peer review of "Diagnosis of Secondary Bacterial Meningitis via Aromatic Metabolites and Biomarkers in Cerebrospinal Fluid"

_ijms, 2025, doi:10.3390/ijms262110522_

Round 1

Reviewer 1 Report

Comments and Suggestions for Authors

Thios work is interesting and well-done, but I think that it coluld be very interesting and important to improving the manuscript if the authors could correlate the presence of determined analytes in CSF with what bacterium has been identifed.

Moreover, it might be important to describe the treatments received by each patient to underline if the presence of a detemrined antibiotic could modify the answers of immune system and produce different analytes, such as IL-6 o S100

Author Response

Reviewer 1

1 ) This work is interesting and well-done, but I think that it could be very interesting and important to improve the manuscript if the authors could correlate the presence of determined analytes in CSF with what bacterium has been identified.

Dear Reviewer,

We sincerely thank you for your time spent studying the article, understanding the complexity of obtaining such rare real samples and valuable comments.

Unfortunately, we do not have information on all pathogens identified in patients with meningitis. However, we inserted some information about microorganisms that were found with cultural methods, revised Table 1 and added the following text in Section 2.1: “Demonstrated microbiological findings were performed for CSF. Among detected bacteria, different nosocomial agents are presented. The presence of Cryptococcus neofor-mans in one sample most likely indicates that there are also bacterial species that were not found with cultural method [https://pmc.ncbi.nlm.nih.gov/articles/PMC6114196/]. Staphyllococcus epidermidis in samples of patients without meningitis is a typical example of contamination. One sample from group 1.1 had positive gram strain but no growth on culture.” 

We also added the following text to a new Section 2.4 “A small number of culture-positive meningitis in group 1 (only 8 patients with a positive CSF culture, Table 1) does not allow us to conduct a correlation analysis between detected bacteria and determined analytes in CSF. However, some information about detected bac-teria and antimicrobial treatment for these four patients will be provided. Patient 1 without secondary meningitis had a surgical intervention for a CNS tumor and had no antibacte-rial treatment. Patient 2 without secondary meningitis had intracranial hemorrhage and received Cefoperason and Sulbactam 2 g + 2 g 2 times per day as antimicrobial prophylax-is. Patient 3 was admitted to our center with sequelae of traumatic brain injury from an-other hospital and then secondary bacterial meningitis occurred. There was no growth in culture and secondary meningitis was diagnosed based on CDC criteria. Antibacterial treatment for secondary meningitis included Meropenem 1 g 3 times per day and Linezol-id 600 g 2 times per day for therapy of nosocomial pneumonia caused by Staphylococcus aureus. Patient 4 with subarachnoid hemorrhage had nosocomial meningitis caused by Klebsiella pneumonia after intracranial pressure monitoring was started. At considered points Imipenem 0.5 g and Cilastatin 0.5 g 4 times per day were started after stopping of Cefoperason and Sulbactam 2 g + 2 g 2 times per day and Amikacin in unknown dosage.”

2) Moreover, it might be important to describe the treatments received by each patient to underline if the presence of a determined antibiotic could modify the answers of immune system and produce different analytes, such as IL-6 o S100

We thank the reviewer for this comment. Unfortunately, we have not enough data on antimicrobial treatment for all patients to conduct any statistical analysis. Nevertheless, we have some data on 4 patients and showed how profiles of metabolites and biomarkers were changed together with leucocyte count in CSF in dynamics for these 4 patients additionally to information about antimicrobial treatment with dosage. This data is explained in a new section 2.4 and demonstrated in a new Fig.1. The text about antimicrobial treatment is inserted to a previous question. The figures and text about dynamics of different parameters including IL-6 in added to a new Section 2.4: “In Fig. 1 a) and b) there are no significant changes over time for the metabolites and biomarkers, which were statistically different in Table 3 (IL-6, p-HPhLA, PhLA, and 3ILA). Even for p-HPhLA on Fig. 1 b) which seems to have dynamic changes with the highest concentration in point 2, its real concentrations were less than 1050 nmol/l. In contrast, Fig. 1 c) demonstrates synchronic dynamics for leucocyte count in CSF and p-HPhLA while IL-6 and two other lactic metabolites did not show significant changes. In Fig. 1 d) p-HPhLA and leucocyte count in CSF change in the same directions, while IL-6 shows opposite dynamics. Since leucocyte count in CSF is commonly used for secondary menin-gitis diagnosis, we compare its dynamics with those for metabolites and biomarkers. Based on obtained data we can conclude that p-HPhLA has the most similar dynamic changes with leucocyte count in CSF.”

Reviewer 2 Report

Comments and Suggestions for Authors

The study analyses secondary bacterial meningitis in different settings, including neurosurgical, and come up with a multivariate model for diagnosis of this condition.

It is astounding that a multivariate model could be built using a  large number of biomarkers and metabolites when having only 53 patients, only 17 of whom had meningitis. As the things stand, the model has to be tested on a larger, preferably different cohort; perhaps bootstrapping would provide more trust in its validity.

There is no information about the etiology – perhaps the difference between the groups are due to different pathogen present. There is little sense in dividing the patients into two groups and the multivariate model, which is the core of the study, combines them together anyway. Why was such a division made?

While there were only 53 patients the authors used  all available samples for analysis – many from the same patient. While it pumps up the numbers it is not correct and some samples were likely from patients already undergoing antibiotic treatment. To illustrate my point - if one had only two patients but had 50 samples from each would a statistical analysis between the two be reasonable?

Other:

Aromatic metabolites of tyrosine, phenylalanine, and tryptophan – what is the source of these – provide more information in the introduction

“patients with sequelae of severe brain damage with a chronic critical illness” – please rewrite as not understandable.

“patients in acute and chronic conditions with and without secondary meningitis” - this description is strange;  what would be acute condition? Neurosurgery? These patients have had underlying condition (tumor), which makes it chronic

There is one patient with CNS infection (meningoencephatitis); why is he/she included as secondary?

Author Response

Reviewer 2

The study analyses secondary bacterial meningitis in different settings, including neurosurgical, and come up with a multivariate model for diagnosis of this condition.

  1. It is astounding that a multivariate model could be built using a  large number of biomarkers and metabolites when having only 53 patients, only 17 of whom had meningitis. As the things stand, the model has to be tested on a larger, preferably different cohort; perhaps bootstrapping would provide more trust in its validity.

Dear Reviewer,

We sincerely thank you for your time spent studying the article, understanding the complexity of obtaining such real samples and valuable comments.

A situation with a number of samples that is less than the number of features is not rare in the clinical studies (the problem “p>>n” https://pmc.ncbi.nlm.nih.gov/articles/PMC10186672/). Nevertheless, in a current study we consider only 8 metabolites and 3 biomarkers for the final model building. For our task it will not be correct to use large models because of the lack of the data. That is why we construct a pretty simple linear model with low risks of overfitting. Also, we used bootstrapping and cross-validation at the patients level to make distribution of the parameters more strict for extracting maximum benefit from our data. This fact was described in Materials and Methods (section 3.3). To avoid further misunderstanding of our article information about this was duplicated in section 2.5. [https://arxiv.org/abs/1811.12808]

2. There is no information about the etiology – perhaps the difference between the groups are due to different pathogen present. 

Unfortunately, we do not have information on all pathogens identified in patients with meningitis. However, we inserted some information about microorganisms that were found with cultural methods, revised Table 1 and added the following text in Section 2.1: “Demonstrated microbiological findings were performed for CSF. Among detected bacteria, different nosocomial agents are presented. The presence of Cryptococcus neofor-mans in one sample most likely indicates that there are also bacterial species that were not found with cultural method [https://pmc.ncbi.nlm.nih.gov/articles/PMC6114196/]. Staphyllococcus epidermidis in samples of patients without meningitis is a typical example of contamination. One sample from group 1.1 had positive gram strain but no growth on culture.” 

We also added the following text to a new Section 2.4 “A small number of culture-positive meningitis in group 1 (only 8 patients with a positive CSF culture, Table 1) does not allow us to conduct a correlation analysis between detected bacteria and determined analytes in CSF. However, some information about detected bac-teria and antimicrobial treatment for these four patients will be provided. Patient 1 without secondary meningitis had a surgical intervention for a CNS tumor and had no antibacte-rial treatment. Patient 2 without secondary meningitis had intracranial hemorrhage and received Cefoperason and Sulbactam 2 g + 2 g 2 times per day as antimicrobial prophylax-is. Patient 3 was admitted to our center with sequelae of traumatic brain injury from an-other hospital and then secondary bacterial meningitis occurred. There was no growth in culture and secondary meningitis was diagnosed based on CDC criteria. Antibacterial treatment for secondary meningitis included Meropenem 1 g 3 times per day and Linezol-id 600 g 2 times per day for therapy of nosocomial pneumonia caused by Staphylococcus aureus. Patient 4 with subarachnoid hemorrhage had nosocomial meningitis caused by Klebsiella pneumonia after intracranial pressure monitoring was started. At considered points Imipenem 0.5 g and Cilastatin 0.5 g 4 times per day were started after stopping of Cefoperason and Sulbactam 2 g + 2 g 2 times per day and Amikacin in unknown dosage.”

3. There is little sense in dividing the patients into two groups and the multivariate model, which is the core of the study, combines them together anyway. Why was such a division made?

We thank the reviewer for this comment. This study included patients with a chronic critical illness (Cohort #1) and post-neurosurgery patients (Cohort #2). In the Cohort #2, patients after neurosurgical intervention developed secondary meningitis within the first week of being in the neurointensive care unit. In Cohort #1 patients were admitted to our intensive care units from other ICUs after the acute phase of their condition had passed, but the patient required an extended stay in the intensive care unit. Such patients in our center are classified as patients with a chronic critical illness and may also subsequently develop secondary meningitis, but usually against the background of an existing primary infectious complication. Thus, in Section 2.2 we planned to analyze the difference of the CSF composition between these two groups. The following text was added in the beginning of Section 2.1 “Patients in Cohort #1 have long-term sequelae of severe brain damage. These patients were admitted to our intensive care units from those in other hospitals after the acute phase of their condition had passed, but the patient required an extended stay in the intensive care unit. These patients are also called patients with a chronic critical illness. This group of patients is characterized by an extended stay in the ICU, in most cases on prolonged mechanical ventilation. Unfortunately, the development of nosocomial infections (pneumonia, secondary meningitis, ventriculitis, urogenital tract infections, and soft tissue infections) is frequent in these patients [19-20]. In Cohort #2, patients after neurosurgical intervention developed secondary meningitis within the first week of being in the neurointensive care unit.” And also the following text was added in the beginning of Section 2.2 “Since patients from cohorts #1 and #2 differ in the time of development of secondary meningitis relative to the primary brain injury, we consider it appropriate to conduct a comparative analysis between  both cohorts #1 and #2 to study the CSF composition in neurosurgery patients and patients with chronic critical illness, as well as between groups within and between cohorts (Table 2).”

4. While there were only 53 patients the authors used  all available samples for analysis – many from the same patient. While it pumps up the numbers it is not correct and some samples were likely from patients already undergoing antibiotic treatment. To illustrate my point - if one had only two patients but had 50 samples from each would a statistical analysis between the two be reasonable?

We thank the reviewer for this comment. We followed patients over time, since it is important to make a diagnostic decision about the presence or absence of meningitis and to monitor the patient's condition at every point. Hence, we included not only points at the time of onset of secondary meningitis but points after antimicrobial treatment starts. Our hypothesis was that metabolites and biomarkers could change dynamically, reflecting the patient's condition, and that these changes would be noticeable in the event of improvement or deterioration. Our hypothesis was based on our previous clinical study on the dynamics of metabolite concentrations in the serum of patients with sepsis, in which we demonstrated not only dynamic changes in metabolite concentrations but also the synchrony of these changes with the clinical picture of the patients' condition. We even observed a slight advance in metabolite changes compared to the SOFA assessment scale [DOI: 10.1134/S106193482470117X]. In the first version of the manuscript, we hesitated to demonstrate these results; however, in the post-review version of the manuscript, we added a new section 2.4 “Dynamic Changes of CSF Parameters in Patients from Cohort #1” in which we provided examples of dynamic changes in metabolites and biomarkers along with changes in the leukocyte count in two patients without meningitis and two patients with meningitis, and also provided available information on the prescribed antimicrobial therapy for these patients. We hope that the added material will be able to demonstrate the validity of our approach to dynamic monitoring of the metabolite and biomarker profile in patients with suspected secondary meningitis.:

“2.4. Dynamic Changes of CSF Parameters in Patients from Cohort #1

We followed patients over time, since it is important to make a diagnostic decision about the presence or absence of meningitis and to monitor the patient's condition at every point. Hence, we included not only points at the time of onset of secondary meningitis but points after antimicrobial treatment starts. Our hypothesis was that metabolites and bi-omarkers could change dynamically, reflecting the patient's condition, and that these changes would be noticeable in the event of improvement or deterioration. Our hypothesis was based on our previous clinical study on the dynamics of metabolite concentrations in the serum of patients with sepsis, in which we demonstrated not only dynamic changes in metabolite concentrations but also the synchrony of these changes with the clinical picture of the patients' condition. We even observed a slight advance in metabolite changes com-pared to the SOFA assessment scale [DOI: 10.1134/S106193482470117X].

To demonstrate the importance of monitoring of the metabolites and biomarkers over time, data for two patients with secondary meningitis and two patients without secondary meningitis from cohort #1 are shown in Fig. 1. A small number of culture-positive menin-gitis in group 1 (only 8 patients with a positive CSF culture, Table 1) does not allow us to conduct a correlation analysis between detected bacteria and determined analytes in CSF. However, some information about detected bacteria and antimicrobial treatment for these four patients will be provided. Patient 1 without secondary meningitis had a surgical in-tervention for a CNS tumor and had no antibacterial treatment. Patient 2 without second-ary meningitis had intracranial hemorrhage and received Cefoperason and Sulbactam 2 g + 2 g 2 times per day as antimicrobial prophylaxis. Patient 3 was admitted to our center with sequelae of traumatic brain injury from another hospital and then secondary bacteri-al meningitis occurred. There was no growth in culture and secondary meningitis was diagnosed based on CDC criteria. Antibacterial treatment for secondary meningitis in-cluded Meropenem 1 g 3 times per day and Linezolid 600 g 2 times per day for therapy of nosocomial pneumonia caused by Staphylococcus aureus. Patient 4 with subarachnoid hemorrhage had nosocomial meningitis caused by Klebsiella pneumonia after intracranial pressure monitoring was started. At considered points Imipenem 0.5 g and Cilastatin 0.5 g 4 times per day were started after stopping of Cefoperason and Sulbactam 2 g + 2 g 2 times per day and Amikacin in unknown dosage.

In Fig. 1 a) and b) there are no significant changes over time for the metabolites and biomarkers, which were statistically different in Table 3 (IL-6, p-HPhLA, PhLA, and 3ILA). Even for p-HPhLA on Fig. 1 b) which seems to have dynamic changes with the highest concentration in point 2, its real concentrations were less than 1050 nmol/l. In contrast, Fig. 1 c) demonstrates synchronic dynamics for leucocyte count in CSF and p-HPhLA while IL-6 and two other lactic metabolites did not show significant changes. In Fig. 1 d) p-HPhLA and leucocyte count in CSF change in the same directions, while IL-6 shows opposite dynamics. Since leucocyte count in CSF is commonly used for secondary menin-gitis diagnosis, we compare its dynamics with those for metabolites and biomarkers. Based on obtained data we can conclude that p-HPhLA has the most similar dynamic changes with leucocyte count in CSF.”

5. Other: Aromatic metabolites of tyrosine, phenylalanine, and tryptophan – what is the source of these – provide more information in the introduction.

We thank the reviewer for this comment. We provided more information in the introduction section: “Severe infectious complications disrupt numerous metabolic pathways in the patient's body. From a diagnostic perspective, altered metabolism of the proteinogenic aro-matic amino acids —tyrosine, phenylalanine, and tryptophan— is of interest, as some metabolites of these amino acids are of microbial origin [9,10], and their profile changes with the development of infectious complications. These metabolites have been studied as prognostic and diagnostic markers of local and systemic infectious processes for more than two decades. In healthy individuals, the source of low concentrations of circulating aromatic microbial metabolites is the intestinal microbiota. With the development of severe infectious complications and sepsis, microbial metabolism of aromatic amino acids changes and occurs not only in the intestine but also at sites of infection. Intermediate or alternative metabolites enter the bloodstream in excess, leading, in particular, to increased levels of circulating sepsis-associated metabolites [11].”

6. “patients with sequelae of severe brain damage with a chronic critical illness” – please rewrite as not understandable.

We thank the reviewer for this comment. Patients with long-term sequelae of severe brain damage were included in our study. These patients were admitted to our intensive care units from other ICUs after the acute phase of their condition had passed, but the patient required an extended stay in the intensive care unit. These patients are also called patients with a chronic critical illness [https://doi.org/10.3390/jcm13133683]. This group of patients is characterized by an extended stay in the ICU, in most cases on prolonged mechanical ventilation. Unfortunately, the development of nosocomial infections (pneumonia, secondary meningitis, ventriculitis, urogenital tract infections, and soft tissue infections) is frequent in these patients. Nosocomial pneumonia caused by Klebsiella pneumoniae occupies a leading position in these patients [https://doi.org/10.17116/anaesthesiology202402139]. This information was added at hte beginning of Section 2.4 “In the present study, two distinct cohorts of patients with suspected secondary men-ingitis were enrolled (Table 1): patients with sequelae of severe brain damage with a chronic critical illness (cohort #1) and patients after neurosurgical interventions (cohort #2). Patients in Cohort #1 have long-term sequelae of severe brain damage. These patients were admitted to our intensive care units from those in other hospitals after the acute phase of their condition had passed, but the patient required an extended stay in the in-tensive care unit. These patients are also called patients with a chronic critical illness. This group of patients is characterized by an extended stay in the ICU, in most cases on pro-longed mechanical ventilation. Unfortunately, the development of nosocomial infections (pneumonia, secondary meningitis, ventriculitis, urogenital tract infections, and soft tissue infections) is frequent in these patients [19-20]. In Cohort #2, patients after neurosurgical intervention developed secondary meningitis within the first week of being in the neuro-intensive care unit.”

7. “patients in acute and chronic conditions with and without secondary meningitis” - this description is strange;  what would be acute condition? Neurosurgery? These patients have had underlying condition (tumor), which makes it chronic

We thank the reviewer for this comment. When we used “patients in acute and chronic conditions” we were considering the rate of infection progression relative to the initial surgical intervention. Thus, in the second cohort, patients after neurosurgical intervention developed secondary meningitis within the first week of being in the neurointensive care unit. In the first cohort, patients were admitted to our intensive care units from other ICUs after the acute phase of their condition had passed, but the patient required an extended stay in the intensive care unit. Such patients in our center are classified as patients with a chronic critical illness and may also subsequently develop secondary meningitis, but usually against the background of an existing primary infectious complication. Since we did not consider the possible interpretation of acute and chronic conditions suggested by Reviewer, we abandoned this distinction throughout the text of the manuscript, replacing it with "post-neurosurgery patients and patients with a chronic critical illness."

8. There is one patient with CNS infection (meningoencephatitis); why is he/she included as secondary?

We thank the reviewer for this comment. This patient is from the Cohort #1 and it means that meningoencephalitis occurred in another ICU as a complication of neurosurgical intervention after severe brain damage. Then, the patient was admitted in our ICU with a diagnosis “secondary bacterial meningoencephalitis”. This is the reason why we consider this patient as a patient with secondary meningitis.

Round 2

Reviewer 1 Report

Comments and Suggestions for Authors

thank you for your revision and for using my suggestions with the aim to improve the article, well-done